# Learning from a Few Shots:
# Data-efficient Cervical Vertebral Maturation Assessment

**Helen Schneider**[*1]               HELEN.SCHNEIDER@IAIS.FRAUNHOFER.DE
**Aditya Parikh**[*1]                ADITYA.PARIKH@IAIS.FRAUNHOFER.DE
**Priya Tomar**[1,2]                  PRIYA.PRIYA@IAIS.FRAUNHOFER.DE
**Maximilian Broß**[1]           MAXIMILIAN.BROSS@IAIS.FRAUNHOFER.DE
**Tom Verhofstadt**[4]                TOM.VERHOFSTADT@MIL.BE
**Anna Konnerman**[3]                KONERMANN@UNI-BONN.DE
**Rafet Sifa**[1,2]                 RAFET.SIFA@IAIS.FRAUNHOFER.DE

[1] *Fraunhofer IAIS*

[2] *University of Bonn*

[3] *University Hospital Bonn*

[4] *Medical Service, Belgian Army*

**Editors:** Accepted for publication at MIDL 2025

## Abstract

The timing of treatment is a crucial decision in orthodontics. Initiating treatment during the appropriate growth phase leads to optimal patient outcomes and can prevent prolonged treatment durations. The most commonly used method for classifying growth phases is cervical vertebral maturation (CVM) assessment, which categorizes CVM into six stages based on the shape and size of the cervical vertebrae. Due to the complexity of manual CVM analysis, it often falls short in performance when assessed visually. Deep learning methods can assist physicians in classifying CVM stages, thus improving orthodontic workflows and treatments. However, a significant challenge in deep learning-based CVM assessment is the limited dataset volume, resulting from difficulties in data collection and annotation. While small training datasets can greatly hinder the model's generalization performance, research on data-efficient training methods for CVM assessment is still lacking. To the best of our knowledge, this paper is the first to evaluate the potential of few-shot learning and in-domain transfer learning for CVM assessment. Specifically, we investigate the architectures ResNet18 and SAM-Med2D. Few-shot learning enhances classification performance by up to 9%. Additionally, in-domain pre-training (using chest X-ray data) results in a significant performance increase of up to 4%.

**Keywords:** Few-shot Learning, Transfer Learning, MedSam, Orthodontic, CVM Assessment

## 1. Introduction

In addition to selecting the appropriate orthodontic treatment, the timing of treatment is crucial for achieving successful outcomes (Liao et al., 2022; Mohammad-Rahimi et al., 2022). The stage of facial growth impacts diagnosis, prognosis, treatment planning, and results. Correctly classifying the growth phase and initiating treatment during the optimal growth period can lead to the best patient outcomes. Conversely, an incorrect classification

---

[*] Contributed equally

may result in prolonged treatment durations or the need for surgical interventions to correct jaw deformities (Mohammad-Rahimi et al., 2022). Several indicators of skeletal maturation have been proposed to assist in determining treatment timing, including dental development and eruption times, hand and wrist maturation, and cervical vertebral maturation (CVM) and morphology. While hand and wrist maturation assessments rely on hand radiographs, CVM utilizes lateral cephalograms, which are commonly used in orthodontic diagnostic procedures (Mohammad-Rahimi et al., 2022). Consequently, CVM assessment is the most frequently employed method by orthodontists, reducing the radiation dose per patient due to the practicality of conventional lateral cephalograms. CVM is classified into six cervical stages (CS1 - CS6) based on the size and shape of the cervical vertebrae (Liao et al., 2022). Previous studies have shown that this classification is highly reliable (Gu and McNamara Jr, 2007; Malta et al., 2009). However, differentiating between these stages in a clinical context can be quite challenging. Variations in clinicians' understanding of skeletal morphology lead to unavoidable subjectivity, which complicates accurate assessment. Consequently, CVM evaluation requires considerable clinical experience and has not achieved satisfactory accuracy when assessed visually (Liao et al., 2022; Chatzigianni and Halazonetis, 2009).

In recent years, computer vision and deep learning (DL) algorithms have demonstrated remarkable performance in the analysis of medical image data (Schneider et al., 2023a, 2024). This potential is also significant in the field of dentistry, where DL-based diagnostic assistance can greatly enhance clinical practices. Studies such as (Tao and Wang, 2022) and (Hwang et al., 2019) have explored various orthodontic applications, including DL-based proximal caries detection. However, cervical vertebral maturation CVM assessment presents a particularly challenging use case for two key reasons. On one hand, ambiguous boundaries between neighboring stages and subjectivity in label annotation can lead to noisy/uncertain labels, which may weaken the classification performance of the DL model (Liao et al., 2022). On the other hand, the limited volume of the data set, resulting from the difficulties associated with data collection, poses a significant challenge for CVM assessment (Liao et al., 2022; Seo et al., 2021). DL methods typically require extensive data sets for supervised training to achieve the remarkable performance necessary for diagnostic decision support. Limited training data can lead to poor generalization behavior of the DL method, hindering their application in clinical worklfows. Due to these challenges data-efficient DL methods are of great interest to researchers, companies, and clinics aiming to develop diagnostic decision support systems for CMV assessment.

## 2. Related Work

To address the challenges faced by physicians, various studies have explored the use of traditional machine learning methods for CVM assessment (Kök et al., 2019; Amasya et al., 2020a,b). The implemented methods rely on handcrafted features, the best performance was achieved with a feed forward neural network. Nevertheless, the application of these traditional machine learning methods in real-world clinical scenarios is constrained by their limited accuracy.

In the recent years DL and computer vision achieved tremendous success in analyzing medical image data (Schneider et al., 2023c,b). In the fields of orthodonics DL algorithms

yielded strong performance among others for abnormality classification in teeth, such as proximal caries detection (Hwang et al., 2019; Lin et al., 2022). Additionally, several studies explored the assessment of CVM using DL algorithms, more precisily convolutional neural networks (CNN), obtaining reasonable accuracy (Hwang et al., 2019; Kim et al., 2021; Seo et al., 2021; Zhou et al., 2021; Mohammad-Rahimi et al., 2022).

However, these studies underline a significant challenge of state-of-the-art CVM assessment: they rely on limited training datasets often fewer than 1,000 samples due to the high costs and time demands of generating extensively annotated data, as well as restricted access to public CVM assessment datasets. These data limitations hinder the performance of supervised DL-based CVM classification.

Despite the severe consequences, research on data-efficient DL methods for cervical vertebral maturation (CVM) assessment is still insufficient. State-of-the-art studies solely focuse on transfer learning based on ImageNet-pretrained weights. However, due to the different varying image modalities, no significant performance improvement has been achieved with ImageNet-based transfer learning (Makaremi et al., 2019).

Few-shot learning (FSL) offers a promising approach for data-efficient training (Wang et al., 2020; Parnami et al., 2022). While remarkable successes have been achieved in various application areas, FSL remains underexplored in the field of dentistry. The authors of (Kim et al., 2024) examined the potential of unsupervised few shot learning for the diagnosis of periodontal disease, highlighting the capability of FSL to address data limitations. However, to the best of our knowledge, FSL has not yet been evaluated for CVM assessment. The aim of our paper is to fill this gap and enable data-efficient CVM classification. Specifically, to the best of our knowledge, our main contributions are:

- First evaluation of FSL for CVM assessment

- Initial examination of in-domain (e.g. utilizing medical images) transfer learning for multi-class and FSL CVM assessment

- First investigation of the foundation model MedSam-2D for CVM classification compared to CNNs for multi-class and FSL training

## 3. Materials and Methods

### 3.1. Dataset

We used the public dataset CVM-900 provided by (Liao et al., 2022). This data set includes 900 clear and distinguishable lateral cephalograms of orthodontic patients aged 7-25 years. The annotation process was conducted through a rigorous multi-expert assessment protocol, where three specialists in orthodontics and radiology independently evaluated all images according to the CVM method developed by (Lamparski, 1975), (Baccetti et al., 2005), and (McNamara and Franchi, 2018). Each expert classified the images into one of six CVM stages based on the morphological characteristics of the second (C2), third (C3), and fourth (C4) cervical vertebrae. Images with unanimous classification were directly included, while those with discrepancies underwent a consensus review session where the experts collectively determined the final classification.

The images were cropped to ensure the cervical vertebrae appeared in a fixed position, reducing interference from other anatomical structures. The images had a resolution of $640 \times 1280$ pixels. For our experiments, we use a subset named CVM-900-Subset introduced in (Liao et al., 2022), which contains 508 samples with clear CVM stages, removing label ambiguity for images that lay on the boundary between two stages. The CVM-900-Subset was divided into training and test sets with an 80-20 ratio using stratified splitting to maintain the label distribution. We resize the images to $256 \times 256$ pixels, to maintain compatibility with the implemented SAM-Med2D architecture. For data augmentation, we applied random horizontal flipping, color jittering, and random rotation within $\pm 30$ degrees.

### 3.2. Model Architecture

We utilize two distinct architectures for the CVM assessment.

**Modified ResNet18** The network is built on the widely-used CNN ResNet (He et al., 2016) known for its impressive performance in medical imaging tasks. Inspired by (Liao et al., 2022), who demonstrated the effectiveness of additional convolutional layers and dropout for CVM assessment, we modified the ResNet18 architecture. We added three convolutional layers after the backbone, incorporating batch normalization and Leaky ReLU activation (alpha=0.1), maintaining the feature dimension to 512. While (Liao et al., 2022) used dropout probabilities of 0.5, we found that a single dropout layer with p=0.3 before the global average pooling and final linear classification layer was sufficient for our task. Both the dropout layer and batch normalization are intended to mitigate overfitting and enhance training stability. Despite the additional convolutional layers, we will refer to this architecture in the following as ResNet18 due to overview reasons. We initialize the ResNet backbone with ImageNet pretrained weights, for the other weights we use Kaiming initialization. Additionally, we aim to evaluate the impact of more effective transfer learning for data-efficient training. To achieve this, we implement pre-training on medical image data, specifically utilizing chest X-ray images from the MedicalMNIST dataset (Yang et al., 2023). In the following sections, we refer to this weight initialization as in-domain transfer learning or Med-ResNet18.

**SAM-Med2D Encoder** The second architecture leverages the image encoder from SAM-Med2D (Cheng et al., 2023), a medical domain adaptation of the Segment Anything Model (SAM) (Kirillov et al., 2023). SAM-Med2D was pre-trained on an extensive medical image dataset, encompassing over 4.6 million images across various modalities. We utilize the Vision Transformer (ViT)-base variant of SAM-Med2D as our backbone encoder. This choice allows us to capitalize on its pre-trained weights, which are specifically tuned for medical imaging tasks, and leverage the powerful transformer architecture. The classification head maintains an identical structure to the ResNet architecture, adapting only the initial input channels from 256 to match the encoder's output. Please note that adapting powerful segmentation foundation models, such as MedSam, for classification tasks is an underexplored research area. This work therefore offers further insights into the adaption of segmentation foundation models for orthodontic use cases.
Both models output logits corresponding to the six CVM stages.

### 3.3. Experiments

Our experiments are conducted using the PyTorch framework on a NVIDIA A100 GPU with 40GB VRAM. To ensure reproducibility, all experiments are executed with three fixed random seeds.

For the ResNet-based models, we employ a batch size of 32 and trained for 100 epochs with early stopping to prevent overfitting with a learning rate of $1e-3$ . For the medical pretrained architectures (SAM-Med2D and Med-ResNet18), we apply a learning rate of $1e-4$ for the encoder, and $1e-3$ for the classification head to further mitigate overfitting. We additionally explored training with both frozen and unfrozen encoder configurations, ultimately selecting an unfrozen approach due to enhanced performance. A learning rate scheduler is implemented to reduce the rate by a factor of 0.1 after 20 epochs. Due to the limited data volume, we initialize the models with the pre-trained ImageNet weights and use the Cross-Entropy (CE) loss function for tradition multi-class (MC) training. We include the training results of a randomly initialized model as baseline into our analysis.

For few-shot learning (FSL) training, an extensive hyperparameter search is performed, evaluating k-shot values of $\{1, 3, 5, 10, 20\}$ and query sizes of $\{5, 10, 20\}$, along with learning rates of $\{1e-3, 1e-4\}$. The training consists of 20 epochs, each with 100 randomly sampled tasks, and model performance is assessed using 50 fixed validation tasks after each epoch. Based on empirical results and computational efficiency, a query size of 10 is chosen for final experiments. The focus for evaluation is on k-shot values of 1, 3, and 5, reflecting realistic clinical scenarios where labeled examples are scarce. Both Binary Cross-Entropy (BCE) and Supervised Contrastive Loss (SCL) are employed to analyze the impact of loss functions. The best models are selected based on the lowest loss on the test set. We evaluate the performance regarding the accuracy, relaxed accuracy and mean absolute error (MAE). The relaxed accuracy considers a prediction as incorrect, when it deviates by more than one class. Due to the ambiguous class boundaries for CVM assessment, the relaxed accuracy and MAE are suitable scores to measure the model performance.

To visualize and interpret our model's focus during predictions, we employ Gradient-weighted Class Activation Mapping (Grad-CAM) using the PyTorch implementation (Gildenblat and contributors, 2021). We apply Grad-CAM to our modified ResNet-18 model's last convolutional layer (layer4) to generate attention maps highlighting regions influential in classification decisions.

## 4. Results

Table 1 shows the highest performance scores from the independent test dataset for the different training methods and architectures. A comprehensive table, considering the k-shot values $\{1, 3, 5\}$, is included in the Appendix (Table 3). The baseline ResNet18 experiments achieve a relatively low accuracy of 58% for traditional MC training. However, the relaxed accuracy of 88% highlights that incorrect predictions typically deviate by only one class. This represents a recognized issue for CVM assesment due to the ambiguous boundaries between neighboring stages and indicates that the model is able to extract the most relevant features. Figure 1 illustrates the t-SNE visualization of high-level features from our ResNet18 model and MC training, showing distinct clusters for the six CVM stages with some overlap. Stages 1 and 6 (earliest and last maturation phases) are the most distinct,

while intermediate stages display gradual transitions, reflecting vertebral development. This emphasizes the challenge of ambiguous boundaries, highlighting the importance of future research on managing ambiguous labels for CVM assessment.

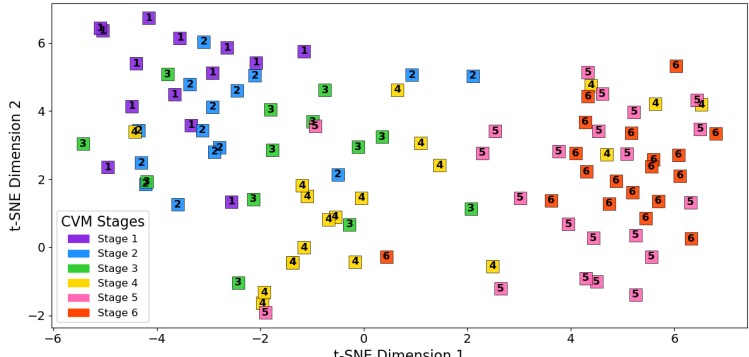

Figure 1: t-SNE visualization of the high-level features of images in CVM-900-Subset. The rectangles in different colors represent samples of different CVM stages.

Additionally, we observed a notable decrease in performance to 47% accuracy when training the ResNet architecture from scratch. This highlights the advantages of transfer learning on out-of-domain data. Further results for the MC and FSL training with random weight initialization can be found in appendix (Table 3). In the following, we organize our results analysis according to the data-efficient methods used.

**Transfer Learning** In-domain transfer learning significantly enhances the performance of the ResNet18 architecture by up to 4% in accuracy for MC training. However, the relaxed accuracy shows only a 1% increase, indicating no significant improvement. This suggests that in-domain transfer learning mainly corrects predictions that initially deviated by just one class for ImageNet pretrained ResNet18 experiments. The more complex ViT architecture of the SAM-Med 2D network leads to lower performance in MC training. Despite being pre-trained on a substantial medical image dataset, we observed a strong tendency for overfitting. Further adapting the SAM-Med 2D model to utilize the strong in-domain pre-training for data-efficient CVM assessment represents future work.

**Few-shot Learning** For the ResNet18 experiments, FSL significantly outperforms MC training. Accuracy increases by up to 9% due to FSL training. The relaxed accuracy of 94% indicates that classes incorrectly classified rarely deviate by more than one class. These enhancement and the MAE reduction of 0.13 further emphasize the improved classification behavior of the model. It is important to note that all these metrics showed significant improvements due to FSL training. These results highlight the advantages of FSL training for CVM assessment, facilitating data-efficient training for orthodontic use case for the ResNet18 experiments.
Despite the significant performance improvements achieved through FSL learning with the ImageNet pre-trained ResNet18, these results are not replicable for the Med-ResNet18 and SAM-Med 2D models. The powerful ViT architecture and/or the use of medical pre-trained

weights lead to overfitting of the limited training data, resulting in no strong performance enhancement. These findings are further visualized in Figure 2, representing the accuracy scores for different k-shot values for all three architectures.

Figure 3 highlights that while SCL outperforms BCE training for all three architectures, only a minimal performance increase is observed. This suggests that the model architecture plays a more crucial role in FSL training than the choice of loss function.

Table 1: Overview of the best ResNet18, Med-ResNet18 and SAM-Med 2D experiments based on the test set. FSL training achieves the highest score for ResNet18, surpassing both MC classification and FSL for the other architectures. Significant differences between the CE training and the FSL methods for one architecture are highlighted with an *. The dagger † represents significant improvement due to in-domain transfer learning.

| Model | Loss | Acc. (%) ↑ | Relaxed Acc. (%) ↑ | MAE ↓ |
|---|---|---|---|---|
| ResNet18 | CE | 53.40 ± 1.59 | 88.03 ± 0.51 | 0.59 ± 0.02 |
|  | BCE (FSL) | 62.14 ± 2.10 * | **94.17 ± 2.38 *** | **0.46 ± 0.06 *** |
|  | SCL (FSL) | **62.46 ± 4.37 *** | 91.59 ± 0.46 * | 0.49 ± 0.03 * |
| Med-ResNet18 | CE | 57.93 ± 0.92 † | 89.32 ± 1.59 | 0.54 ± 0.03† |
|  | BCE (FSL) | 56.31 ± 2.10 | 89.00 ± 0.92 | 0.57 ± 0.02 |
|  | SCL (FSL) | 56.96 ± 3.57 | 91.26 ± 0.79 | 0.54 ± 0.05 |
| SAM-Med 2D | CE | 48.87 ± 1.65 | 87.06 ± 0.92 | 0.67 ± 0.02 |
|  | BCE (FSL) | 47.90 ± 2.78 | 85.44 ± 0.79 | 0.68 ± 0.03 |
|  | SCL (FSL) | 49.19 ± 5.16 | 86.41 ± 0.79 | 0.66 ± 0.06 * |

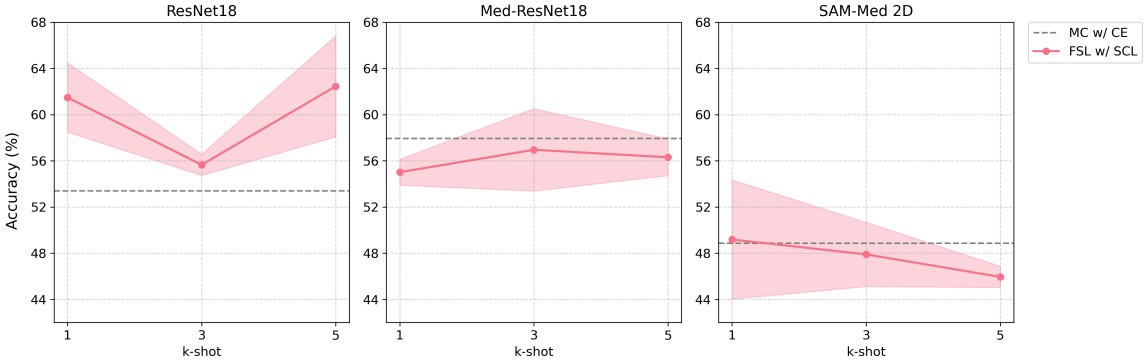

Figure 2: The accuracy of FSL training (red) and MC baselines (grey) across ResNet18, Med-ResNet18, and SAM-Med 2D architectures, with k-shot values $k \in \{1, 3, 5\}$. MC performance peaks with Med-ResNet18, showing benefits of in-domain transfer learning. However, FSL training achieves highest accuracy with ResNet18 for most k-shot values, indicating no advantage from in-domain transfer learning for FSL. SAM-Med 2D shows lowest performance across all settings. We observe no clear trends for k-shot values. Overall, FSL training surpasses in-domain transfer learning for data-efficient CVM assessment.

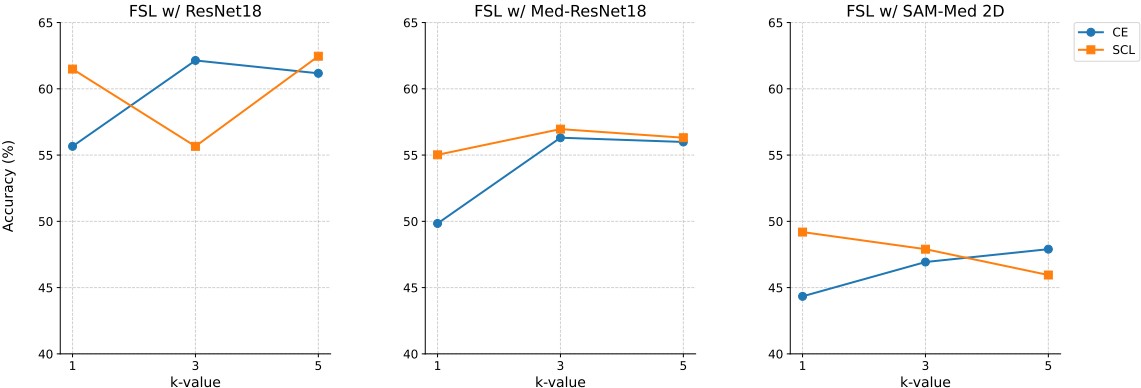

Figure 3: The accuracy for FSL training of the SCL (orange) and the BCE (blue) loss across ResNet18, Med-ResNet18, and SAM-Med 2D architectures, with k-shot values $k \in \{1, 3, 5\}$. For each architecture, the highest performance is achieved with the SCL loss. The SCL loss surpasses the BCE loss seven out of nine experiments. However, only a minimal difference is observed between the loss functions regarding the highest accuracy scores for each architecture.

In addition to the performance evaluation, we analyze the interpretability of the preferred FSL method. Figure 4 visualizes five patient samples. For the accurate prediction of CS6 (fourth sample), the heat map highlights the high relevance of the cervical vertebrae C2, C3, and C4, indicating a well-informed model decision. We observe a similar correct focus for the patient in CS4 (sixth sample). Conversely, for the incorrect prognosis of class number 1 (first and second sample), the model primarily focuses on the patient's jaw, resulting in an unreliable decision. For the incorrect prediction of the third sample, the model doesn't focus on the highly relevant cervical vertebrae C2, C3, and C4. Although we observed this behavior for several test samples, future work includes an extensive interpretability analysis.

## 5. Conclusion

In conclusion, the analysis of various data-efficient training methods and architectures demonstrates that while in-domain transfer learning improves the performance of the ResNet18 model, its impact is limited, especially regarding relaxed accuracy. The results show that FSL training significantly surpasses both MC training and transfer learning for ResNet18, highlighting its potential for data-efficient strategies in CVM assessment. However, despite being pretrained on a large medical image dataset, the state-of-the-art SAM-Med 2D network did not facilitate data-efficient training for either MC or FSL. Future work will concentrate on addressing the challenges posed by limited datasets and ambiguous label boundaries for CVM assessment by integrating data-efficient FSL with label distribution learning methods.

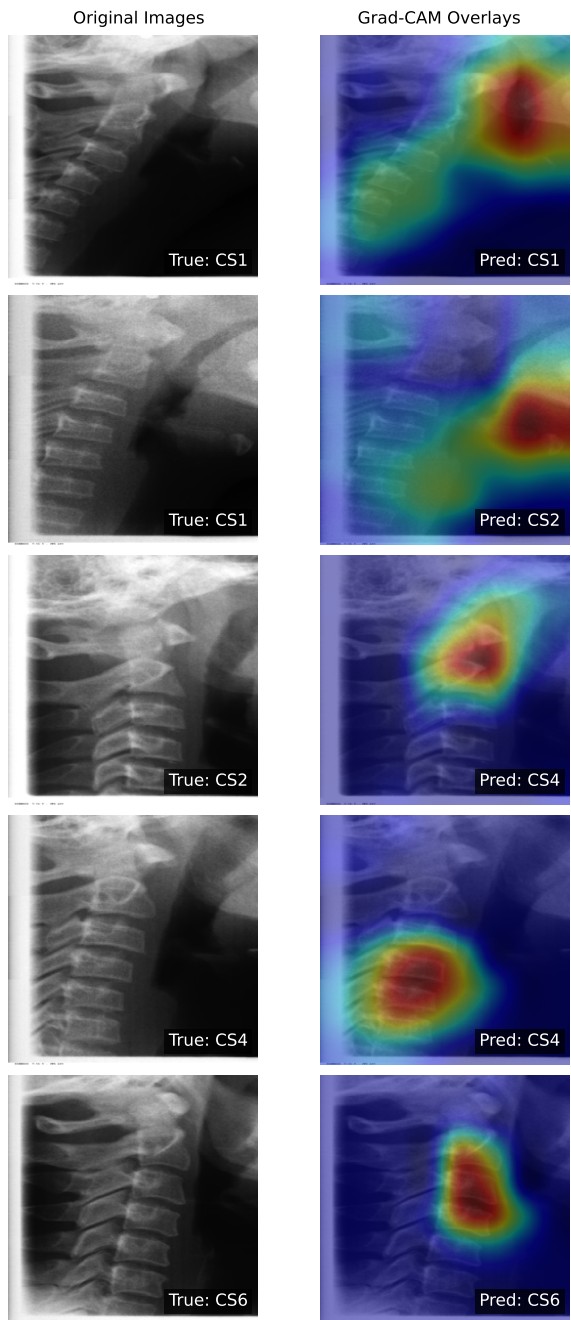

Figure 4: Grad-CAM visualizations showing model attention: The first three rows demonstrate incorrect predictions with diffused and misplaced attention, while the subsequent two rows show correct predictions with focused attention on relevant vertebral regions, where red indicates regions of highest importance for the model's decision.

## Acknowledgments

This research has been funded by the Federal Ministry of Education and Research of Germany and the state of North-Rhine Westphalia as part of the Lamarr-Institute for Machine Learning and Artificial Intelligence, LAMARR22B.

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

## Appendix A. Detailed Result Table

Table 2: Class-wise performance of different models using various loss functions (CE, BCE, SCL) across six classes (CS1 to CS6).

| Model | Loss | CS1 | CS2 | CS3 | CS4 | CS5 | CS6 |
|---|---|---|---|---|---|---|---|
| Resnet18 | CE | $69.05 \pm 13.47$ | $42.22 \pm 11.33$ | $28.21 \pm 15.81$ | $66.67 \pm 6.24$ | $50.00 \pm 7.42$ | $57.89 \pm 21.49$ |
|  | BCE | $66.67 \pm 3.37$ | $57.78 \pm 3.14$ | $43.59 \pm 3.63$ | $61.67 \pm 2.36$ | $60.61 \pm 4.29$ | $71.93 \pm 4.96$ |
|  | SCL | $61.90 \pm 12.14$ | $57.78 \pm 8.31$ | $48.72 \pm 7.25$ | $66.67 \pm 2.36$ | $68.18 \pm 0.00$ | $64.91 \pm 13.13$ |
| Med-ResNet18 | CE | $64.29 \pm 10.10$ | $35.56 \pm 11.33$ | $30.77 \pm 10.88$ | $65.00 \pm 7.07$ | $60.61 \pm 9.34$ | $56.14 \pm 4.96$ |
| SAM-Med 2D | CE | $69.05 \pm 3.37$ | $26.67 \pm 10.89$ | $38.46 \pm 6.28$ | $55.00 \pm 4.08$ | $37.88 \pm 2.14$ | $64.91 \pm 4.96$ |

Table 3: Performance comparison of different models and FSL methods for k-shot values $k \in \{1, 3, 5\}$.

| Model | k | Acc. (%) | | Relaxed Acc. (%) | | MAE | |
| --- | --- | --- | --- | --- | --- | --- | --- |
| | | BCE | SCL | BCE | SCL | BCE | SCL |
| Resnet18 (Scratch) | - | 46.93 ± 0.46 | - | 83.88 ± 1.27 | - | 0.80 ± 0.06 | - |
| FSL w/ Resnet18 (Scratch) | 1 | 41.75 ± 0.00 | - | 87.38 ± 0.00 | - | 0.71 ± 0.00 | - |
| | 3 | 47.57 ± 0.00 | - | 86.41 ± 0.00 | - | 0.71 ± 0.00 | - |
| | 5 | 51.46 ± 0.00 | - | 90.29 ± 0.00 | - | 0.64 ± 0.00 | - |
| Resnet18 | - | 53.40 ± 1.59 | - | 88.03 ± 0.51 | - | 0.59 ± 0.02 | - |
| FSL w/ Resnet18 | 1 | 55.66 ± 3.00 | 61.49 ± 3.00 | 92.56 ± 1.21 | 93.53 ± 1.65 | 0.53 ± 0.04 | 0.47 ± 0.02 |
| | 3 | 62.14 ± 2.10 | 55.66 ± 0.92 | 94.17 ± 2.38 | 93.85 ± 1.65 | 0.46 ± 0.06 | 0.51 ± 0.03 |
| | 5 | 61.17 ± 2.10 | 62.46 ± 4.37 | 92.88 ± 1.21 | 91.59 ± 0.46 | 0.47 ± 0.01 | 0.49 ± 0.03 |
| Med-Resnet18 | - | 57.93 ± 0.92 | - | 89.32 ± 1.59 | - | 0.54 ± 0.03 | - |
| FSL w/ Med-Resnet18 | 1 | 49.84 ± 3.99 | 55.02 ± 1.21 | 89.32 ± 3.63 | 90.61 ± 0.92 | 0.62 ± 0.05 | 0.56 ± 0.02 |
| | 3 | 56.31 ± 2.10 | 56.96 ± 3.57 | 89.00 ± 0.92 | 91.26 ± 0.79 | 0.57 ± 0.02 | 0.54 ± 0.05 |
| | 5 | 55.99 ± 0.92 | 56.31 ± 1.59 | 92.23 ± 1.59 | 90.94 ± 2.29 | 0.53 ± 0.03 | 0.55 ± 0.03 |
| SAM-Med 2D | - | 48.87 ± 1.65 | - | 87.06 ± 0.92 | - | 0.67 ± 0.02 | - |
| FSL w/ SAM-Med 2D | 1 | 44.34 ± 1.21 | 49.19 ± 5.16 | 86.41 ± 2.86 | 86.41 ± 0.79 | 0.71 ± 0.02 | 0.66 ± 0.06 |
| | 3 | 46.93 ± 0.92 | 47.90 ± 2.78 | 85.76 ± 2.29 | 86.41 ± 1.59 | 0.71 ± 0.03 | 0.67 ± 0.04 |
| | 5 | 47.90 ± 2.78 | 45.95 ± 0.92 | 85.44 ± 0.79 | 85.11 ± 1.21 | 0.68 ± 0.03 | 0.71 ± 0.03 |

