# OpenReview forum: "Learning from a Few Shots: Data-efficient Cervical Vertebral Maturation Assessment"
_MIDL.io/2025/Conference — MIDL 2025 Poster_

### Official Review · Reviewer_yJuJ · 2025-02-09

**Confidence:** 4
**Preliminary Rating:** 4
**Recommendation:** Poster

**Summary:**

Using ResNet18 and SAM-Med2D models, the authors test these methods on a small dataset of X-ray images. FSL boosts ResNet18's accuracy, making it better than standard training methods. However, SAM-Med2D, despite being pre-trained on a large medical dataset, struggles due to overfitting. The study shows FSL is a promising tool for medical tasks with limited data, though more work is needed to adapt advanced models like SAM-Med2D for small datasets.

**Strengths:**

The study introduces few-shot learning (FSL) for cervical vertebral maturation (CVM) assessment, improving accuracy with limited data. It demonstrates FSL's effectiveness with ResNet18, offering a practical solution for small datasets in orthodontics. The exploration of in-domain transfer learning and SAM-Med2D adds novelty, though SAM-Med2D's overfitting highlights room for improvement.

**Weaknesses:**

1) Limited dataset size (CVM-900-Subset) restricts generalizability. 2) Overfitting issues with complex models like SAM-Med2D and Med-ResNet18, especially in FSL scenarios. 3) Lack of extensive interpretability analysis, particularly for incorrect predictions.

**Detailed Comments:**

The study introduces few-shot learning (FSL) for cervical vertebral maturation (CVM) assessment, improving accuracy with limited data. It demonstrates FSL's effectiveness with ResNet18, offering a practical solution for small datasets in orthodontics. The exploration of in-domain transfer learning and SAM-Med2D adds novelty, though SAM-Med2D's overfitting highlights room for improvement.

**Justification Of The Preliminary Rating:**

The paper is well-structured, with a clear introduction that outlines the importance of CVM assessment and the challenges associated with it. The authors provide a thorough review of related work, highlighting the limitations of existing methods and the potential of deep learning in this domain. The methodology is detailed and well-explained, with a focus on two architectures: a modified ResNet18 and the SAM-Med2D encoder.

**Questions To Address In The Rebuttal:**

1) Limited dataset size (CVM-900-Subset) restricts generalizability. 2) Overfitting issues with complex models like SAM-Med2D and Med-ResNet18, especially in FSL scenarios. 3) Lack of extensive interpretability analysis, particularly for incorrect predictions.

**Special Issue:**

No

---

> ### Author Response · Authors · 2025-03-07
>
> Dear reviewer,
>
> thank you a lot for your helpful feedback!
>
> 1. Weakness limited dataset size
> For the CVM assessment problem, only limited public data is available, and due to the niche nature of this issue, current state-of-the-art research has used small training datasets without utilizing data-efficient machine learning methods. While extensive training data can significantly enhance performance, it is currently unavailable to the broader research community. Additionally, we do not anticipate the publication of a comprehensive CVM assessment datasets, as is typical in areas like chest X-ray analysis, due to the specialized nature of CVM assessment. Thus, data-efficient CVM assessment represents a critical research question in the field of orthodontic research. We therefore present the first investigation of the potential of few shot learning and in-domain transfer learning for CVM assessment, powerful data-efficient methods.
>
> 2. Overfitting
> We implemented standard methods to prevent overfitting, such as dropout layers, partially training the later layers of the networks (which are responsible for more complex feature generation), and adapting the learning rate. Additionally, we incorporated an adaptation of the ResNet model proposed in publication (Liao, N. et al. iCVM: An Interpretable Deep Learning Model for CVM Assessment Under Label Uncertainty. IEEE journal of biomedical and health informatics) to address overfitting in CVM assessment. We explored the potential of data-efficient methods, including few-shot learning and transfer learning, to further reduce overfitting. The analysis of enhanced data-efficient methods for CVM assessment, such as updating the SAM architecture to mitigate overfitting, will be part of our future work.
>
> 3. Interpretability Analysis
> To deepen the interpretability analysis we included additional samples, underlining the correct focus on the general vertebra area for correct predictions and mislead focus for incorrect ones.
>
> We hope these explanations and updates are beneficial!

---

### Official Review · Reviewer_Sf87 · 2025-02-17

**Confidence:** 4
**Preliminary Rating:** 3
**Final Rating:** 4

**Summary:**

This paper proposes a deep learning model trained in a few-shot setting for CVM maturation classification to address the limited data problem. The model utilizes ResNet18 as the main architecture and SAM-Med2D encoder for comparison. Accuracy, BCE and SCL are utilized to evaluate the model's performance.

**Strengths:**

1. The problem investigated leads to wide potential applications.
2. The proposed model seems to have good performance, and the Grad-CAM visualization demonstrates the model’s robustness.
3. Utilizing few-shot methods to address the data shortage problem is a good idea.

**Weaknesses:**

1. Experiments. The model's performance seems to be good; however, there are only three models compared. It would be nice to see more comparisons for different encoders or other widely used medical imaging architectures (e.g., UNet) to further demonstrate the model’s effectiveness. Also, adding more details about datasets and the model's performance on individual classes can provide more comprehensive comparisons.
2. Some parts of the introduction could benefit from further addressing: the authors noted that ‘ambiguous boundaries between neighboring stages and subjectivity in label annotation can lead to noisy/uncertain labels, which may weaken the classification performance of the DL model.’. It seems that in the experimental setting, the ambiguous labels are removed. It would be good to see the comparison between ambiguous labels removed or not removed and how the models perform on ‘ambiguous’ data.

**Detailed Comments:**

Minor error: There is a typo in the introduction section ‘worklfows’, please correct me if I am wrong.

**Justification Of The Final Rating:**

Through adding detailed experiments, the authors demonstrate the effectiveness of the proposed method across categories, and conducting few-shot learning is worthwhile for the medical AI research field in small-size real-world datasets.

**Justification Of The Preliminary Rating:**

Overall, the problem and the method proposed have wide application value, and it is good to see that few-shot learning has the potential to leverage the data shortage problem as they claimed. However, the authors could provide more thorough comparisons to demonstrate their model's effectiveness, which makes me keep my current rating.

**Questions To Address In The Rebuttal:**

Please refer to the weakness part.

---

> ### Author Response · Authors · 2025-03-07
>
> Dear Reviewer,
>
> thank you for your helpful feedback!
>
> 1. Weakness Experiments:
> We included an overview of the classwise performance for an enhanced generalization analysis in the appendic. We included a more detailed description for the utilized data set.
> While it is true that we compared only three different models, we also investigated 3 different training methods/loss functions per model, and conducted a statistical analysis which requires several training runs for one experiment to evaluate the statistical significance of each solution approach. If we fined suitable pre-trained classification networks as additional baselines we will include it till the camera-ready baseline. But since we are currently working with a extremely limited data set, we fined the basic comparison between state-of-the-art convolutional neural network and vision transformer sufficient.
>
> 2. Ambiguous label experiments
> We intentionally excluded ambiguous labels from this study as we focus on data-efficient deep learning training rather than noise-robust training. The implemented methods, such as transfer learning and few-shot learning, are not suitable for noise-robust training. Including noisy annotations would make it difficult to determine whether the observed overfitting is due to limited training data or noise overfitting, complicating the performance analysis of the data-efficient machine learning methods. However, in future work, we aim to address the challenge of ambiguous labels alongside data-efficient training (e.g. by introducing few-shot learning for ordinal labels) and will include both ambiguous and non-ambiguous labels. To further underline the importance of ambiguous labels as future work, we included a tsne visualizationof high-levek features for the resnet model. While we observe distinct clusters per class, gradual transitions in between classes is observed.
>
> We hope these explanations and updates are beneficial!

---

> > ### Comment · Reviewer_Sf87 · 2025-03-11
> >
> > Dear authors, I have read all of your responses and your revised manuscript. The updated experiment results strengthen the proposed method, and it is worth while to develop few-shot methods in small-size datasets. Therefore, I raise my score to 4.

---

### Official Review · Reviewer_cjpE · 2025-02-21

**Confidence:** 4
**Preliminary Rating:** 3
**Final Rating:** 3

**Summary:**

The authors tackle the task of 'growth phase' estimation from cervical vertebrae images in xray shots, that are used in orthodontics. This seems a niche but important task in the timing of performing orthodontic treatments.
Because it is a niche task there are only small datasets available, and therefore the authors focus on applying and evaluating transfer learning and few-shots learning, and their efficacy on such small datasets.
They compare a Resnet initialized from imagenet weights, a resnet pretrained on medical data, and a SAM network.
These are compared in two regimes, 'regular' classification learning, 'few shot' learning where a hyper parameter search is used to determine various settings, the latter method is compared with two losses.

**Strengths:**

The authors tackle a real life situation of solving an important niche task with limited data.
This makes the paper more applied in nature, but also serves as a realistic scenario.

Additionally model-attention is visualized, which is a nice addition.

**Weaknesses:**

It is hard to determine how effective the actual approach of few shot learning is due to some issues.
- The few-shot learning approach has a parameter search, thus it gets a lot more 'tries' to get a good score and the best result is selected against the _test_ set, which creates a clear bias towards better results.
- The main baseline is a resnet trained from imagenet features in the basic multi class learning approach, it would be nice to have a fully trained baseline to see how that compares. Additionally since this is a niche task there is no global baseline to compare against; this is inherent due to the niche task but still makes it harder to assess if the baseline is a fair comparison.

**Detailed Comments:**

The acronyms can be repeated a bit more often, in the figures especially. It's kinda annoying for the reader to figure out where the acronym is mentioned first and figure out what FSL and MC are in this context.

**Justification Of The Final Rating:**

Thanks for the authors to updating the paper and improve it!
I'm still at a 3, but mostly because of the small datasize making it hard to assess in the end how this method would generalize and reducing the scope of the paper. Still it's a nice applied paper.
Btw, the updated paper seems to have a page too many, but that could be the styling with the highlighting causing that.

**Justification Of The Preliminary Rating:**

The method targets an interesting niche problem that seems to be important, but the paper has some methodological issues. When addressing a niche problem, it is harder to determine fair comparisons (since there are few or none to compare against) and this scrutiny becomes stricter.

**Questions To Address In The Rebuttal:**

Especially address the methodological issue where the FSL has it's best run selected against the test set during the hyper parameter search.

---

> ### Author Response · Authors · 2025-03-07
>
> Dear Reviewer,
>
> We want to thank you for your helpful feedback!
>
> 1. Weakness: hyperparameter search and data split:
> Only 508 samples are available in the public dataset with clean labels. Initial experiments included generating train, validation, and test sets; however, reasonable splits resulted in insufficient training data volume for model training. We therefore only focused on train and test data. However, we plan to further investigate our methodology on clincial in-house data, enabling us to work with sufficient train, valid, and test data.
> We did not conduct an extensive hyperparameter search for the multiclass training because we based most of our hyperparameters on the publication (Liao, N. et al. iCVM: An Interpretable Deep Learning Model for CVM Assessment Under Label Uncertainty. IEEE journal of biomedical and health informatics), which uses the same backbone architecture and data set for CVM assessment. Due to this prior work the hyperparameters of the multiclass training needed less tuning. However, since our paper is the first to investigate few shot learning for CVM assessment, hyperparameter tuning was crucial, as we could not rely on other publications for reasonable settings.
>
> 2. Weakness: model training from scratch:
> We did not train the model from scratch because the publication (Liao, N. et al. iCVM: An Interpretable Deep Learning Model for CVM Assessment Under Label Uncertainty. IEEE journal of biomedical and health informatics) underlined that imagenet pretraining can be beneficial to mitigate the overfitting tendencies. However, we run the corresponding experiments, multi-class training achieves a significant performance drop regarding the accuracy when trained from scratch. We will include the results in the paper as additional information. Please note that for few-shot learning we provided the results for one-seed training. We will add the mean and std. for several seed training for the camera ready version, since our experiments are not finished yet.
>
> We hope these explanations and updates are beneficial!

---

### Author Rebuttal · Authors · 2025-03-07

**Rebuttal:**

Updated manuscript with described changes highlighted in yellow!

**Supporting Material:**

/attachment/a07266a9c78aac76b8aa590485e3442e3cf915b2.pdf

---

### Comment · Area_Chair_jgCw · 2025-03-09
**Discussion Period**

Dear Reviewers,​

Thanks for your time and effort in reviewing this paper. This is the right time to discuss this paper with each other.​

The authors have provided a rebuttal to your comments and uploaded a revision. Please review their responses and the revised manuscript. For the preliminary recommendation, we have two borderlines and one weak accept.​ Considering the authors' responses and the discussion, please update your rating and assessments for the paper.

Any discussion is welcome, and you may consider reading each other's reviews, posting questions for clarification, and reaching a consensus.​

Best,
Your AC

---

### Comment · Area_Chair_jgCw · 2025-03-14
**Urgent discussion due in about one day**

Dear all the Reviewers,

The discussion period is nearing its conclusion. Please update your final rating with justification if you haven't already. Two reviewers voted to accept this paper, while one reviewer considered it borderline. In my view, this paper could be accepted. Any discussion is welcome!

Best,
Your AC

---

### Meta-Review · Area_Chair_jgCw · 2025-03-22

**Recommendation:** Accept (Poster)
**Confidence:** 3

**Metareview:**

While the reviewers raised issues about limited dataset size and lack of comparisons, on balance the proposed few-shot learning has some practical value. There is still a lot of room for improvement. The scope of the paper is limited though. I think it would still make a good conference contribution.